# Glycaemic Imbalances in Seizures and Epilepsy of Paediatric Age: A Literature Review

**DOI:** 10.3390/jcm12072580

**Published:** 2023-03-29

**Authors:** Emanuele Bartolini, Anna Rita Ferrari, Simona Fiori, Stefania Della Vecchia

**Affiliations:** 1Department of Developmental Neuroscience, IRCCS Stella Maris Foundation, 56128 Pisa, Italyannarita.ferrari@fsm.unipi.it (A.R.F.);; 2Tuscany PhD Programme in Neurosciences, 50139 Florence, Italy; 3Department of Clinical and Experimental Medicine, University of Pisa, 56128 Pisa, Italy; 4Department of Molecular Medicine and Neurogenetics, IRCCS Stella Maris Foundation, 56128 Pisa, Italy

**Keywords:** epilepsy, hypoglycaemic seizures, hypoglycaemia, hyperglycaemia, diabetes mellitus, paediatric age

## Abstract

Cerebral excitability and systemic metabolic balance are closely interconnected. Energy supply to neurons depends critically on glucose, whose fluctuations can promote immediate hyperexcitability resulting in acute symptomatic seizures. On the other hand, chronic disorders of sugar metabolism (e.g., diabetes mellitus) are often associated with long-term epilepsy. In this paper, we aim to review the existing knowledge on the association between acute and chronic glycaemic imbalances (hyper- and hypoglycaemia) with seizures and epilepsy, especially in the developing brain, focusing on clinical and instrumental features in order to optimize the care of children and adolescents and prevent the development of chronic neurological conditions in young patients.

## 1. Introduction

The interplay between blood sugar levels and susceptibility to seizures is especially complex. Glucose is the main energy supply of the central nervous system. The human brain accounts for only 2% of body weight but consumes about 20% of glucose-derived energy of the whole body, and cerebral metabolic requests are likely much higher in paediatric age [1,2,3,4,5]. This remarkable metabolic demand is due to both neuronal workflow (generation of action and postsynaptic potentials, maintenance of ion gradients, and resting potentials) and the biosynthesis of neurotransmitters by neurons and astrocytes [2,3,4]. The grey matter utilizes a significantly greater amount of energy compared to the white matter [5,6], and the demand for glucose briskly increases with neuronal activation [7].

Glycogen stores in the brain are tiny and limited to astrocytes, thus the brain is reliant on a continuous intake of glucose from the systemic circulation. Glucose movements within different compartments happen through glucose transporters (GLUTs). The entry within the central nervous system is mediated by the GLUT1 subtype, which allows facilitated diffusion through the blood–brain barrier. GLUT1 also mediates glucose uptake from brain extracellular fluid into glial cells. Instead, the GLUT3 subtype lets glucose flow into neurons [2] and is much more efficient than GLUT1, insomuch as neurons are privileged with respect to glial cells in case of high metabolic demand [2].

In human cells, energy can be produced from glucose by glycolysis in the cytosol and by oxidative phosphorylation (oxphos) in mitochondria. Intracellular glucose is initially metabolized to pyruvate by glycolysis, with no request for oxygen. Thereafter, pyruvate enters the mitochondrion and undergoes oxphos, which is much more efficient than glycolysis in terms of energy production; oxphos can only be performed in aerobiosis. Pyruvate is, instead, transformed to lactate in anaerobiosis, and energy production as ATP molecules is only obtained by low-efficiency glycolysis. The healthy brain may increase both glycolysis and oxphos in order to maximize the energy supply after acute activation [8].

In epilepsy, there is a derailment of glucose catabolic pathways. Seizures greatly enhance the cerebral metabolic rate, increasing oxygen consumption, cerebral blood flow, and glucose uptake by neurons. Glucose metabolism is acutely shifted towards glycolysis and lactate production (ictal hypermetabolism), followed by a postictal decrease below baseline (postictal hypometabolism) [9,10]. Mitochondrial oxygen consumption also increases acutely, yet there is a net shift towards less efficient glycolysis despite aerobiotic conditions, reminiscent of the Warburg effect observed in cancer cells. Cerebral glucose availability may also be limited, because, in chronic epilepsy, GLUT transporters may be dysfunctional [10,11].

On the other side, the disruption of mitochondrial oxphos could be involved in epileptogenesis. Experimental oxphos inhibition results in the destabilization of hippocampal membrane potentials and provokes epileptiform activity in initially healthy male rats [12]. Neuronal excitability can also be directly affected by glycaemic levels. In the animal model, blood glucose concentrations positively correlate with susceptibility to seizures, and diabetes mellitus favours blood–brain barrier alterations during experimental epileptic seizures [1,13]. In humans, both hyper- and hypoglycaemic conditions have been found to exacerbate seizures [14,15,16]. As a matter of fact, glucose imbalances influence the brittle energy homeostasis of the brain. A disruption of energy availability affects the sodium–potassium pump and the resting state potential and increases intracellular calcium and reactive oxygen species that promote cell death [17]. Hyperglycaemia can directly increase neuronal excitability acting on the ATP-sensitive potassium channels of hippocampal and neocortical neurons; hypoglycaemia depresses GABA levels enhancing excitatory transmission [18,19].

Seizures usually improve with the control of glycaemic status in patients with type 1 diabetes mellitus (T1DM) and type 2 diabetes mellitus (T2DM) [20], whereas fluctuations in blood glucose have been associated with drug-resistant epilepsy [21].

In this paper, we aim to review the existing knowledge on the association between acute and chronic glycaemic imbalances (hyper- and hypoglycaemia) with seizures and epilepsy, especially focusing on the developing brain of children and adolescents. The main issues we will deal with are summarized in Figure 1.

## 2. Materials and Methods

The literature review was performed by searching PubMed and Scopus search engines for full papers in English up to and including April 2022 (10 April 2022). We used the keywords: (hypoglycem*) AND (epilep* OR seizur*) AND (child* OR paediatric population) (n = 1377 + 179), (hyperglycem* OR non-ketotic hyperglycem* OR diabetic ketoacidosis OR hyperosmolar hyperglycaemic state) AND (epilep* OR seizur*) AND (child* OR paediatric population) (n = 183 + 35), (diabet* mellitus) AND (epilep* OR seizur*) AND (child* OR paediatric population) (n = 571 + 389). Zotero 5.0 reference manager was used to exclude duplicate papers. The full text of all potentially eligible articles and their supplementary information were obtained and independently assessed by two authors (E.B. and S.D.V.). We resolved any ambiguities regarding eligibility through discussion.

## 3. Results

### 3.1. Hypoglycaemia

Hypoglycaemia is a condition characterised by the lowering of plasma glucose levels. Although there is no uniform cut-off, a plasma glucose level of 50 mg/dL or less has been considered sufficient to define hypoglycaemia, as many counterregulatory responses occur at this level [22,23,24]. Severe hypoglycaemia is a medical emergency presenting with neuroglycopenic (e.g., confusion, impairment of vigilance, behaviour disturbances, acute symptomatic seizures) and autonomic symptoms (e.g., diaphoresis, tachycardia, tremulousness) [25]. Acute symptomatic seizures develop with different characteristics according to the age of onset and aetiology of hypoglycaemia, and reflect aberrant neurotransmitter metabolism, cerebral blood flow, and even blood–brain barrier and microvascular function [26]. Hypoglycaemia can acutely induce the release of excitatory amino acids, such as glutamate, resulting in neuronal hyperexcitability [26], chronically reducing the availability of cerebral glycogen stored by astrocytes [26,27]. It can also lead to structural brain abnormalities that may persist even after the hypoglycaemia has resolved, favouring the development of neurocognitive deficits and epilepsy in the long term [28,29].

#### 3.1.1. Neonatal Hypoglycaemia

In paediatric populations, hypoglycaemia is not a rare condition. Its incidence reaches a peak in the new-born (0–28 days of age) and decreases with age [30,31]. Healthy neonates can experience transient hypoglycaemia as a part of the normal adaption to extrauterine life in the first 24–48 h after birth [32]; glucose levels gradually increase to reach adult values (>70 mg/dL) within the first 72–96 h (transitive-adaptive hypoglycaemia) [33,34]. Severe and prolonged lowering of blood glycaemia can instead promote acute neurological symptoms and require urgent treatment. Neonatal hypoglycaemia can depend on inadequate glycogen stores or transient secondary hyperinsulinism at birth [35]. The deficiency of glycogen stores occurs through excessive anaerobic glycolysis in prematurity, because of placental insufficiency, and perinatal asphyxia [36]. Hyperinsulinism-related hypoglycaemia can be observed in children of diabetic mothers due to intrauterine exposure to elevated blood glucose levels [23,35,37,38]. In metabolic disorders, seizures can be precipitated by incidental hypoglycaemia, yet low blood sugar levels and seizures can also be different and co-occurring phenotypical expressions [39].

Neonatal hypoglycaemia can be immediately followed by acute seizures [40] or lead to long-term epilepsy due to structural brain damage [41,42,43]. Predictors of neurological sequelae (i.e., epilepsy, intellectual disability, and focal neurological deficits) are somatic comorbidities (i.e., fever and cardiac/respiratory failure), prolonged acute seizures, and brain lesions on neuroimaging, regardless of the aetiology of the hypoglycaemia [40].

Neuroglycopenic symptoms in newborns can be nonspecific and include feeding difficulties, irritability, and hypotonia. The semiology of seizures in this age range is often very subtle. Although neonatal seizures can be classified as motor (automatisms, clonic, epileptic spasms, myoclonic, and tonic) and nonmotor (autonomic seizures and behavioural arrest), their features may not be overt and overlap with nonepileptic clinical phenomena, especially in the intensive care setting (e.g., jittery, tremors, and movement disorders). Many newborns will have mostly electrographic-only seizures, detectable exclusively by polygraphic video-EEG or amplitude-integrated EEG (aEEG) [44]. Hypoglycaemia-induced seizures cannot be distinguished from those provoked by other aetiologies.

Neuroimaging can disclose structural brain abnormalities, define their extent, and help to define the prognosis. Neonatal transcranial ultrasound has low sensitivity [45]. Conversely, magnetic resonance imaging (MRI) can accurately identify acute lesions that typically affect the parietal and occipital lobes, with a predilection for the white matter [42,46,47]. Diffusion restriction in these areas can be acutely identified from the first days of life, either unilaterally or bilaterally [48]. The involvement of basal ganglia/thalamus, peri-rolandic regions or pyramidal tracts and middle cerebral artery infarctions can also be seen [47,49]; hippocampal sclerosis can be identified in the long term [50]. The early detection of brain structural damage predicts an unfavourable neurodevelopmental outcome [51]. The affected regions will evolve in cystic encephalomalacia, atrophy, and gliosis, in turn promoting the development of chronic epilepsy. The reason for the frequent involvement of the parietal and occipital lobes is not exactly known [52,53]. Some authors have hypothesized that those regions are targeted due to their abundance in neuronal migration/proliferation, synaptogenesis, excitatory neurotransmission, and intense metabolic activity during the neonatal period [42,54,55,56,57,58]. Accordingly, basal ganglia and peri-rolandic regions could also be targeted in view of their high metabolic demand.

An example of brain damage associated with neonatal hypoglycaemia and resulting in symptomatic epilepsy is provided in Figure 2 (personal observation).

The characteristics of hypoglycaemia-provoked epilepsy have been especially described in low-income countries, wherein hypoglycaemia is not rare due to the limited availably of routine blood glucose monitoring at birth [30,59]. Kapoor at al. [60] retrospectively described a cohort of children with chronic epilepsy and severe neurologic deficits (global developmental delay in 91.2%) after neonatal hypoglycaemia, showing that the most common were epileptic spasms (76.4%) and focal seizures with visual aura (11.2%), respectively, prevailing in younger and older children. Parieto–occipital brain lesions were observed in the entire cohort, including those with epileptic spasms only. Drug-resistance was observed in about 70% of cases. Those with epileptic spasms commonly developed hypsarrhythmia and a full-blown West syndrome; Lennox–Gastaut syndrome (LGS) (2.4%) and continuous spikes and waves during sleep (CSWS) (1.2%) were also described [60]. Other studies have confirmed the possible late development of LGS and CSWS, either arising from earlier West syndrome or independently [42,54,56,60]. The frequent occurrence of epileptic spasms and focal parieto–occipital seizures has been widely confirmed [42,46,50,54,56,60,61].

Table 1 summarizes epilepsy, EEG, and MRI features in published studies on children with seizures/epilepsy associated with neonatal hypoglycaemia.

It has been proposed that basal ganglia damage and leukomalacia would promote epileptic spasms, whilst isolated parieto–occipital damage would mostly be associated with focal seizures [64,65,66,67]. The electro-clinical features of the latter can mimic self-limited epilepsy with autonomic symptoms (impaired awareness with paroxysmal eye phenomena such as blinking, clonic movements, tonic deviations, and eyelid fluttering), yet ictal vomiting is rare [46,54,66,68,69] (personal observation in Figure 3). These patients often also exhibit visual loss/agnosia with abnormal visual-evoked potentials to compose a complex neurological phenotype defined by Karimzadeh et al. as ‘Hypoglycaemia–Occipital Syndrome’. Patients may need rehabilitation in order to improve their visual impairment [46].

Myoclonic seizures have also been sporadically reported in patients with peri-rolandic brain lesions [40].

Epileptic spasms tend to be more drug-resistant than focal seizures with visual aura, possibly confirming a more severe underlying pathology [42,54]. The severity of epilepsy would also depend on the patient’s age. Episodes of status epilepticus have been mostly described in younger children, whilst epilepsy can wane in older patients [34,50].

Some children with neonatal hypoglycaemia should be specially monitored for episodes of severe recurrent hypoglycaemia that can relapse several times from birth to childhood in specific syndromes driven by hyperinsulinism (e.g., congenital hyperinsulinism, Beckwith–Wiedmann syndrome, Soto syndrome), insufficient energy supply (i.e., inborn errors of metabolism that result in deficiencies in glycogen, amino acids, or free fatty acids), or deficiency in cortisol or growth hormone (e.g., Costello syndrome, hypopituitarism, congenital adrenal hyperplasia) [23,40,70]. Focusing on children with inborn errors of metabolism, Gataullina et al. [40] reported that only half of the patients who suffer seizures during the first hypoglycaemic event (n = 90/170; 53%) can experience further seizures later on (n = 23/90; 23%). The first hypoglycaemic seizure could either be self-limited (68%) or very prolonged to establish an overt status epilepticus (32%). Children with status epilepticus at the onset would be at risk of further prolonged episodes, especially triggered by fever. In this series, most children had symptomatic epilepsy with brain lesions on MRI (86%), whose pattern depended on age at the onset of hypoglycaemia: posterior white matter (0–6 months), basal ganglia (6–22 months), parieto–temporal cortex (>22 months). A minority of patients (14%) developed recurrent hypoglycaemic seizures followed by non-hypoglycaemic seizures despite normal neuroimaging [40].

#### 3.1.2. Hypoglycaemia in Older Children

Although hypoglycaemia decreases in frequency as one moves away from the neonatal period, a brisk lowering of glucose levels can also occur and promote seizures in older patients. We have already mentioned children may still develop severe recurrent hypoglycaemia. Hypoglycaemic events may also arise *de novo* in childhood [25,71]. The spectrum of underlying causes is broad, including primitive or secondary hyperinsulinism, metabolic disorders, and iatrogenic hypoglycaemia.

Seizures have been especially described in preschoolers with T1DM who experience hypoglycaemia due to excessive insulin intake [72]. On the other hand, hypoglycaemia has been found in 0.5% of nondiabetic children with seizures [73]; this figure is likely an underestimation as hypoglycaemia might be overshadowed by postictal glucose increases induced by endogenous corticosteroid release [74]. The pathophysiology of seizures induced by hypoglycaemia is unclear. In animal models, acute hypoglycaemia triggers epileptiform discharges in the amygdala and hippocampus [75]. In humans, most reported seizures in hypoglycaemia are convulsive. Nevertheless, Lapenta et al. [76] described a 61-year-old diabetic patient in whom insulin-induced transient hypoglycaemia triggered an EEG-documented temporal lobe seizure, yet with a semiology different from typical temporal lobe epilepsy—nocturnal convulsions rather than focal impaired awareness seizures with epigastric aura and oromotor automatisms. In children, most studies report a predominance of nocturnal convulsions alike [72]. The predilection for the nighttime depends on low blood sugar levels that may pass un-noticed and persist for more than 2–4 h during sleep [77]. Children may also exhibit hemiclonic, tonic, and myoclonic seizures, especially those with inborn errors of metabolism [40]. We may hypothesize that hypoglycaemia targets both the mesial temporal lobe and the peri-rolandic cortex, which has a high metabolic demand and possibly favours a motor presentation of seizures. Hypoglycaemia in childhood per se would not favour the development of epilepsy in the long term unless it provokes structural brain damage—especially after status epilepticus—or belongs to wider epileptogenic phenotypes such as inborn errors of metabolism [40,78]. This rarely happens outside of the neonatal phase.

On the other hand, persistent hypoglycaemia in children with malformations of cortical development should alert physicians to suspect pathogenic variants in the PI3K-AKT-mTOR pathway. These individuals may exhibit segmental overgrowth, drug-resistant seizures, and a continuum of brain malformations spanning from megalencephaly to focal cortical dysplasia. The PI3K-AKT-mTOR pathway hyperactivation leads to increased intracellular glucose uptake and reduced hepatic gluconeogenesis, with net hypoglycaemia in the absence of serum insulin. The resulting hypoinsulinaemic, hypoketotic hypoglycaemia can be rescued by glucose infusion [79,80]. It is still unknown whether avoiding hypoglycaemia in these patients may ameliorate the seizure frequency.

In the developing brain, a low blood sugar level modifies the brain’s electrical activity. In school-aged children, glycaemia below 41.4 mg/dL (2.3 mmol/L) shifts the EEG background activity from the alpha to theta band, whilst restoring normoglycaemia reverses to the alpha band [81]. Recurrent severe hypoglycaemias in children would not change the EEG background activity in the long term [82] but instead promote permanent focal abnormalities (focal/generalized slowing or epileptiform discharges) [83]. It is unknown whether these children might develop recurring unprovoked seizures (i.e., epilepsy) in adulthood.

Of note, patients with inborn errors of metabolism who suffer episodes of neonatal-infantile hypoglycaemia can also develop features of idiopathic generalized epilepsy in childhood. The few reported cases all exhibit primary hyperinsulinaemic hypoglycaemia (i.e., excessive insulin production due to known or presumed genetic defects). Gataullina et al. [40] reported three patients with episodes of severe recurrent hypoglycaemia during the first 2 years of life who later developed features of self-limited epilepsy with centrotemporal spikes or early-onset absence epilepsy, successfully treated by valproate. Two of these patients harboured mutations in SUR1, encoding a potassium channel [84]. On the other hand, Descamps et al. reported a case of hyperinsulinaemic hypoglycaemia arising at 15 years of age in a boy with a former diagnosis of idiopathic generalized epilepsy; remarkably, tonic–clonic seizures were flanked by unusual ‘red flags’ such as the absence of status epilepticus with vegetative symptoms and facial myoclonia. Diazoxide treatment allowed the resolution of both glycogenic symptoms and seizures, resulting in a complete withdrawal of antiepileptic medications [85]. Another case of hyperinsulinaemic hypoglycaemia with apparently generalised epilepsy has been reported in a 2-year-old girl harbouring a mutation of *GCK*—a gene encoding glucokinase. In this case, antiseizure medications, octreotide, and diazoxide had no positive effects, whereas starting a ketogenic diet resulted in the resolution of neuroglycopenic signs and seizure freedom [86].

#### 3.1.3. Cerebral Hypoglycaemia

An inadequate supply of glucose to neurons may also occur in patients with normal glycaemia. As previously mentioned, the central nervous system relies on specific transporters that facilitate glucose diffusion from systemic to brain circulation. Mutations of the gene encoding for the GLUT1 transporter (*SLC2A1*) hamper the passive diffusion of glucose through the blood–brain barrier and result in a fully neurological phenotype named De Vivo/GLUT1 deficiency syndrome (Glut1DS) [87]. The phenotype derives from the overlap of a continual progressive dysfunction (microcephaly/deceleration of head growth, intellectual disability, and ataxia) and paroxysmal symptoms (eye–head movement abnormalities, seizures, and movement disorders). Expressivity is variable. Patients may develop only a part of these symptoms, which indeed tend to occur at different ages: paroxysmal eye–head movements and seizures in early infancy, developmental impairment, ataxia, paroxysmal exertion-induced dystonia, and further movement disorders later on until adolescent/adult age. Young children may exhibit isolated early-onset absence seizures (<4 years of age), to be considered a Glut1DS red flag [88,89]. The diagnosis can be obtained by the combination of clinical signs, findings form lumbar puncture analysis performed after a four- to six-hour fast (hypoglycorrhachia) and molecular analysis (pathogenic *SLC2A1* variants) [88]. To suspect and obtain an expedite diagnosis is of paramount importance. In Glut1DS, a ketogenic diet is the first-line treatment and should be initiated as early as possible to provide a prompt supplemental supply of metabolic fuel from ketone bodies to the developing brain. The ketogenic diet can also be beneficial for other patients with drug-resistant epilepsy, irrespective of the underlying cause, especially in the paediatric age. It can be continued indefinitely but might be poorly tolerated by adolescents and adults. The low glycaemic index treatment (LGIT) and the modified Atkins diet (MAD) are diets recently introduced for refractory epilepsy. MAD and LGIT have antiepileptic efficacy with fewer side effects compared to ketogenic diets. The MAD uses high fat, low carbohydrate, moderate protein diet to induce ketosis [88,90]. The LGIT stabilizes blood glucose instead of increasing ketone bodies, allowing the intake of a limited amount of carbohydrates (40–60 g/day); the percentage of calories from fat is about 60%, compared with up to 90% in the ketogenic diet [91].

#### 3.1.4. Differential Diagnosis between Hypoglycaemic Events and Seizures

Distinguishing hypoglycaemic and epileptic events may not be straightforward. Neuroglycopenic symptoms (e.g., confusional state, erratic behaviour, and autonomic disturbances) may overlap with features of focal impaired awareness seizures. Hypoglycaemia can also show up with motor phenomena such as eye twitching or limb jerks [92,93,94,95,96,97,98]. As mentioned previously, acute symptomatic convulsive seizures may arise during hypoglycaemia, sometimes leading to an inappropriate diagnosis of epilepsy.

A common misdiagnosis regards patients with insulinoma, whose EEG can show focal/diffuse slowing during the event but also focal spikes and sharp waves can be detected [96,99]. This tumour is very rare in paediatric age and must be considered when apparently refractory seizures relapse in children with macrosomy relative to the familial constitution [100]. Interestingly, Jaladyan and Darbinyan [98] have described a 13-year-old girl treated with multiple antiseizure medications for myoclonia and convulsions on awakening; she was initially misdiagnosed with juvenile myoclonic epilepsy before an insulinoma was detected. As generally happens in hypoglycaemic crises provoked by insulinoma, this girl experienced symptoms after fasting, especially in the morning or in the late afternoon [101].

Such diagnostic misinterpretations can lead to enduring pseudoresistance, continuously switching ASMs that are completely ineffective, for seizure mimics such as hypoglycaemic events.

A challenging differential diagnosis regards nonconvulsive status epilepticus. When neuroglycopenic symptoms are prolonged, the EEG can show slow rhythmic abnormalities that can be remitted after benzodiazepine administration, partially fulfilling the diagnostic criteria for nonconvulsive status epilepticus. A diagnostic clue is a dissociation between an improved EEG and the persistence of altered vigilance as long as glycaemia is not restored [102,103]. Eventually, gathering valuable clinical information at the onset, duration, and type of clinical manifestations is fundamental. If the episodes are prolonged, occur mainly in fasting situations or in connection with antidiabetic oral therapy intake, and improve after food, one should preferentially lean towards a diagnosis of hypoglycaemic crisis [99,102].

### 3.2. Hyperglycaemia

Hyperglycaemia can affect different age groups and be sustained by variable aetiologies. The mechanisms by which it induces seizures are largely unknown. Lowering the seizure threshold due to increased GABA metabolism, cerebrovascular dysfunction, neuronal hyperosmolarity, and dehydration can all play a role in provoking acute symptomatic seizures [74,104,105,106].

#### 3.2.1. Neonatal Hyperglycaemia

Hyperglycaemia is rarer than hypoglycaemia among newborns, yet an expedite identification is fundamental to limit the high mortality and morbidity it brings in this age group. Neonatal hyperglycaemia is defined by a serum glucose level higher than 150 mg/dL (8.3 mmol/L) or whole blood glucose above 125 mg/dL (6.9 mmol/L), regardless of gestational or postmenstrual age. The safe range is usually considered to be 70–150 mg/dL [107]. The causes of neonatal hyperglycaemia span from iatrogenic factors (maternal use of diazoxide, neonatal intake of caffeine, steroids, phenytoin, or inappropriate parenteral glucose treatment), the inability to metabolize glucose (prematurity, intrauterine growth restriction, diabetic mother), to stressful events that increase endogenous glucocorticoids (sepsis, pain, hypoxia, and respiratory distress) [107]. Seizures would especially occur after brisk glucose elevation in predisposed individuals, such as preterm infants who have a decreased ability to suppress endogenous glucose production, immature insulin response to glucose, and limited glycogen and fat stores. Bruns et al. [108] described a cohort of newborns and children with severe iatrogenic hyperglycaemia; in their study, the occurrence of seizure was inevitably associated with coma and unfavourable neurological outcome. The seizure semiology in these children has not been systematically addressed. Nevertheless, as discussed beforehand, neonatal seizures are often subtle, independently from the underlying aetiology, with motor and nonmotor features. Hyperglycaemia peaks also negatively affect seizure control in neonates with established epileptic encephalopathies, who are, per se, exposed to a brittle glucose homeostasis [109].

#### 3.2.2. Hyperglycaemia in Older Children

After the neonatal period, diabetes mellitus largely plays a major role. Sudden glycaemia elevation can especially result in diabetic ketoacidosis (DKA) and a hyperosmolar hyperglycaemic state (HHS). These conditions are alarming complications that can arise abruptly at diabetes onset or during the disease course, triggered by a brisk imbalance between the effects of insulin and that of counterregulatory hormones (i.e., glucagon, catecholamines, cortisol, and growth hormone), often due to intercurrent disorders. In DKA, absolute or relative insulin deficiency prevents cellular glucose intake, promotes gluconeogenesis, activates lipolysis to release glycerol and free fatty acid, increases ketone bodies production, and leads to metabolic acidosis. In HHS, insulin deficiency is milder and metabolic derangement is progressive with the development of hyperglycaemia (plasma glucose level > 33.3 mmol/L or 600 mg/dL), dehydration, and plasma hyperosmolality (serum osmolality > 320 mmol/kg) over several days in the absence of appreciable metabolic acidosis and ketonemia [110]. The term nonketotic hyperglycaemia (NKH) is often used to include situations in which an HHS state is not strictly fulfilled.

In adults, HHS may provoke a generalized brain dysfunction leading to a state of coma and also focal neurological signs such as hemichorea/hemiballism or acute symptomatic seizures [110,111]. Seizures occur mostly as epilepsia partialis continua (EPC), i.e., recurrent and unrelenting hemiclonic seizures originated by the peri-rolandic cortex. Visual/imperceptive and prolonged aphasic seizures have also been reported in decreasing order of frequency [112,113,114,115,116,117]. Diabetic children may similarly develop HHS, even though more rarely than adults, and suffer unilateral focal motor seizures and EPC [118,119,120,121,122]. This type of recurring seizure mandates glycaemia to be restored in order to obtain seizure freedom. Chronic epilepsy may residue in some cases [123,124]. Like adults, children may also exhibit hemichorea/hemiballismus, in most cases mutually exclusive with seizures [125,126]. The outcome of seizures in children with HHS is usually favourable when an expedite metabolic treatment is established and brain lesions are prevented, even though there are exceptions. A worsening epilepsy trajectory has been described in a 3-year-old child, who developed continuous spike and waves in sleep two years later than diabetes-related EPC [123].

The role of DKA in seizure development is conflicting and not completely clear. In general, DKA is deemed to promote seizures less often than HHS. Ketone bodies, indeed, exert an antiepileptogenic effect, which is exploited by the ketogenic diet in some forms of epilepsy (e.g., GLUT1 deficiency) [115,127]. However, in HHS, seizures can be provoked by ketosis-induced hyperventilation/hypocapnia or cerebral venous thrombosis due to coagulation derangement. The treatment of ketosis itself can result in side effects that promote seizures: life-threatening brain oedema, electrolyte imbalances, and excessive glycaemia lowering [128,129]. Focal motor seizures have been anecdotally described in newborns [130] and more often in school-aged children and adolescents with DKA as an expression of diabetes decompensation. It is self-intuitive that patients who develop acute brain damage are exposed to chronic epilepsy. From the neurophysiological standpoint, the EEG background activity slows in children with DKA irrespective of seizure occurrence, according to the levels of serum glucose and ketone bodies but independently of the pH [131]. Shober et al. [132] also reported that children with epilepsy and T1DM are especially prone to develop DKA, suggesting a bidirectional association between seizures and ketoacidosis whose underlying pathophysiology is still obscure.

Neuroimaging plays a pivotal role to diagnose and monitor the outcome of seizures both in HHS and DKA. In adults with HHS, the following brain MRI hallmarks have been reported over the brain regions from which seizures originate: cortical T2W hyperintensity, restricted diffusion and faint enhancement, and subcortical T2W hypointensity with a ‘negative shine-through’ effect [114]. No specific early radiological signs of DKA have been described. Hippocampal vasogenic oedema followed by hippocampal sclerosis and epilepsy in the long term has been described in single cases of HHS and DKA in the long term [124,133]. Brain MRI can reliably identify severe complications of both conditions such as cerebral oedema, especially using diffusion-weighted imaging [134], focal infarctions typically occurring in the basal ganglia, thalamus and brainstem’s grey matter [135,136,137], cerebral herniations [137,138], and extrapontine myelinolysis [139]. Cerebral oedema would be more common in DKA compared to HHS, likely due to the absence of hypocapnia in the latter [140], whilst the risk of thrombosis would be greater in HHS in view of the pronounced hypertonicity [141,142].

#### 3.2.3. Diabetes Mellitus and Epilepsy Comorbidity

Seizures may occur in people with diabetes as a comorbid condition rather than a consequence of acute hyperglycaemia. Such comorbidity is bidirectional, irrespective of diabetes and epilepsy subtype [143]. Diabetes mellitus can be defined by fasting plasma glycaemia above 125 mg/dl but also by additional criteria such as random plasma glucose of ≥200 mg/dL in patients with symptoms of hyperglycaemia or hyperglycaemic crisis (Appendix A). Epileptogenic microscopic/macroscopic brain lesions can be induced by glycaemia fluctuations, and diabetes may be affected by the metabolic side effects of antiseizure medications. Autoimmune factors and genetic background are also involved [28,144,145,146,147].

Several studies have demonstrated a particular T1DM/epilepsy association. T1DM is considered to be due to autoimmune pancreatic β-cell destruction. The resulting absolute insulin deficiency usually shows up in childhood and youngsters, and may also arise in senior patients as slowly progressive insulin-dependent diabetes mellitus (SPIDDM) and latent autoimmune diabetes of adulthood (LADA). Autoimmune markers of T1DM include islet cell autoantibodies and autoantibodies to GAD (GAD65), insulin, the tyrosine phosphatases IA-2 and IA-2β, and zinc transporter 8 (ZnT8) [148].

Diabetic children develop epilepsy 2-3 times more commonly than normoglycaemic controls [144,145,149,150]. Younger age at diabetes onset is the main risk factor [144]. A clear association of T1DM with incident idiopathic generalised epilepsy has been reported in children and young adults [151,152]. A direct role of autoimmunity has been proposed to drive the comorbidity T1DM-epilepsy. Anti-glutamic acid decarboxylase antibodies (GAD-Ab) are known to trigger T1DM, especially those targeting the 65 kDa isoform (GAD65-Ab). These are especially linked to the subtypes of insulin-dependent diabetes mellitus (SPIDDM) and latent autoimmune diabetes in adults (LADA), often misdiagnosed as T2DM. High GAD65-Ab levels (at least 100-fold) have been found in autoimmune neurological disorders (e.g., stiff-person syndrome, cerebellar ataxia, and limbic encephalitis), sporadic drug-resistant temporal lobe epilepsy in children and adults, and in the subgroup of patients with T1DM who develop seizures. Low levels might promote both T1DM and seizures, even though further studies are needed to support this hypothesis [153,154,155].

In adults, T2DM is acquiring a growing prevalence in diabetic patients and the comorbidity with epilepsy is weaker, even though still consistent. T2DM has not an autoimmune trigger but results from the progressive loss of adequate β-cell insulin secretion frequently on the background of insulin resistance [148]. Amongst senior adults, those with drug-resistant epilepsy exhibit a two-fold increased prevalence of diabetes—both the less prevalent T1DM and the more common T2DM, mostly linked to cryptogenic/unknown and symptomatic epilepsy especially due to atherothrombosis damage. As in children, the onset of diabetes mostly precedes the development of epilepsy (80%). Adults with diabetes, including those with T1DM, tend to develop focal epilepsies commonly arising from the temporal lobe [156].

Epilepsy and diabetes can also share a common molecular background [156]. A complex genetic landscape is likely implicated in most cases. However, unique shared genetic defects can be responsible for the comorbidity in monogenic diabetic syndromes that escape the dichotomy T1DM/T2DM [148]. Syndromes characterized by neonatal-onset diabetes and early-onset epilepsy are described (mutations in *GCK*, *INS*, *KCNJ11*, and *ABCC8*) [157]. In our experience, we observed an interesting case of comorbidity between epilepsy and maturity-onset diabetes of the young (MODY). The MODY phenotype encompasses a group of autosomal-dominant monogenic disorders with the non-insulin-dependent form of diabetes, classically presenting in adolescence or young adults before the age of 25 years. In Figure 4, we show how long-term video-EEG revealed typical absence seizures in a young patient with MODY due to a pathogenic mutation of the glucokinase gene (*GCK*), also carrying GAD autoantibodies. Mutations of GCK promote the MODY2 subtype, which is characterized by mild stable fasting hyperglycaemia [158]. A minority of patients (about 1%) with MODY may exhibit positive GAD antibodies, whose role is unclear [159].

This anecdotal observation, never reported beforehand, suggests genetic background and autoimmunity might affect the propensity to those seizures usually regarded as genetic-driven, such as idiopathic genetic epilepsy (namely childhood absence epilepsy) (Figure 4).

It should also be mentioned that stress hyperglycaemia (transient high blood glucose levels with spontaneous resolution after the acute illness regresses) [160] has a high prevalence in children with febrile convulsions, especially after prolonged febrile convulsions, and predicts further episodes [161].

The interplay between DM and epilepsy is further confirmed by recent evidence about the effects of antidiabetic drugs on seizures. Metformin has shown promising results as an antiepileptic agent in many experimental studies. In vivo, it has been successfully employed to reduce seizures in people with tuberous sclerosis complex and Lafora disease (reviewed in [162]). Rosiglitazone can suppress seizures in vitro, hampering the presynaptic glutamate release in hippocampal slices [163]. These preliminary data may pave the way to the repositioning of antidiabetic drugs for epilepsy treatment.

## 4. Conclusions

Brain excitability and systemic metabolic balance are strictly intermingled. The energy supply to the neurons critically depends on glucose, whose fluctuations can promote immediate hyperexcitability resulting in acute symptomatic seizures. Hypoglycaemia is particularly epileptogenic, especially in newborns who have high metabolic demands, in children with neurometabolic disorders, and in those with brittle glucose homeostasis due to diabetes mellitus. An expedite recognition of hypoglycaemic symptoms prompt the investigation of the underlying cause and prevents brain damage, which in turn yield chronic epilepsy. Hyperglycaemia provokes acute symptomatic seizures less often, mostly when acute complications of diabetes mellitus ensue. Instead, diabetes mellitus is remarkably linked to chronic epilepsy and such comorbidity should be carefully investigated. Defining homogenous cut-offs in children for hypoglycaemia/hyperglycaemia-induced brain damage is complicated, and clinical observation is strikingly important. The complex interplay between glycaemia and seizure susceptibility must always be considered in the developing age, in order to optimize the care of children and prevent the development of chronic neurological conditions in young patients.

## Figures and Tables

**Figure 1 jcm-12-02580-f001:**
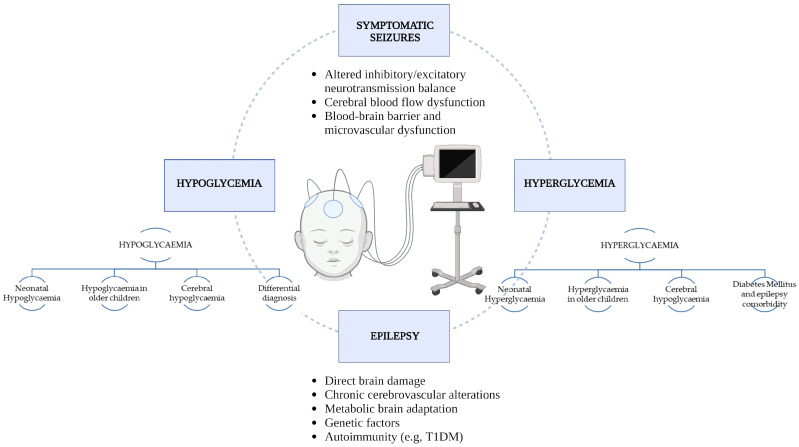
Flow chart summarizing the main themes of the review and the potential mechanisms implicated in the pathophysiology of epilepsy in the context of hypo- and hyperglycaemia.

**Figure 2 jcm-12-02580-f002:**
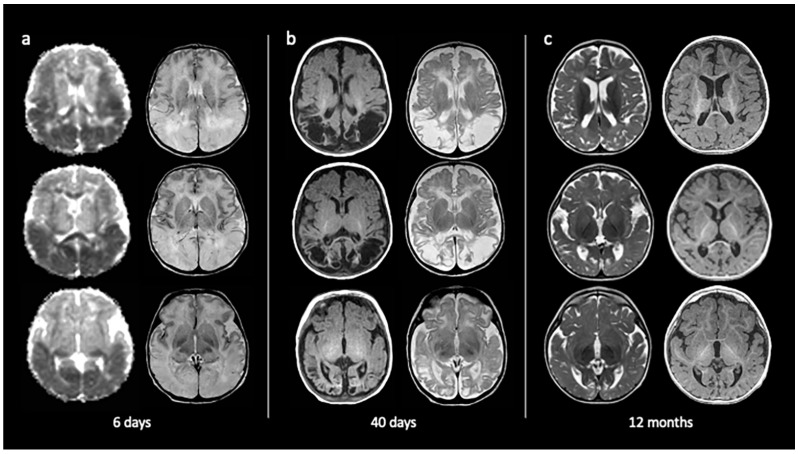
Evolving brain injury in brain MRI from perinatal hypoglycaemia. Symptomatic hypoglycaemia was detected on day 4, postnatally. A posterior brain lesion pattern is reported here, with predominant parietal and occipital lobe involvement. From the left to the right side of the images, brain MR axial images of the same patient are shown at 6, 40 days, and 12 months of age. In detail, (**a**) DWI (left) and T2W (right) images showing abnormal signal intensities and oedema in the parietal and occipital regions; (**b**) T1W (left) and T2W (right) images showing the evolution of hypoglycaemia-related abnormalities to brain poromalacia in the same parietal and occipital regions; (**c**) T2W (left) and T1W (right) images depict chronic brain abnormalities with parietal and occipital white and grey matter volume reduction and local signal tissue abnormalities, in particular involving occipital cortical areas.

**Figure 3 jcm-12-02580-f003:**
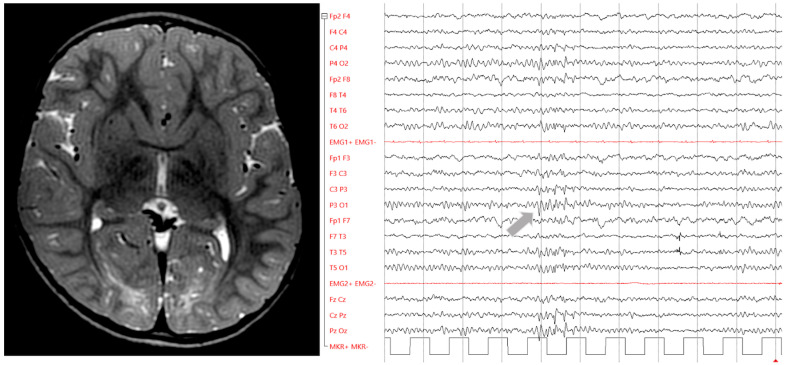
Brain MRI and EEG in focal symptomatic epilepsy in perinatal hypoglycaemia. Axial FLAIR 1.5T brain MRI (**left**) depicts bilateral symmetrical hyperintensity of the occipital white matter in a 12-year-old child who suffered recurrent convulsive seizures during neonatal hypoglycaemia and still exhibits drug-resistant focal impaired awareness seizures with staring, sialorrhea and left eye/head deviation. EEG recording (10–20 International System) (**right**) display sharp waves in a burst over the posterior brain areas, predominating on the left parieto–occipital region (arrow).

**Figure 4 jcm-12-02580-f004:**
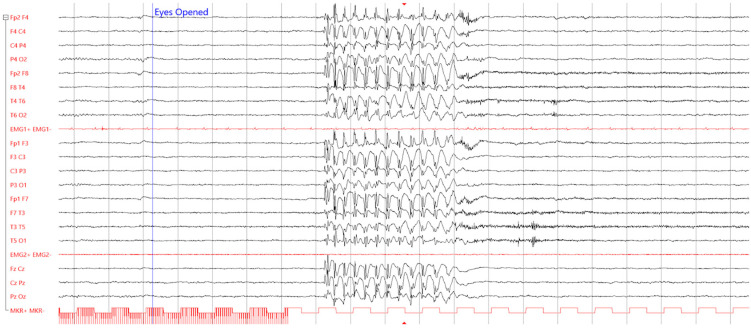
Example of comorbidity between young-onset diabetes mellitus and idiopathic generalized epilepsy. EEG recording (10–20 International System) of a 10-year-old boy affected by MODY (inherited CGK mutation) referred for recurrent episodes of unresponsiveness. The exam displays diffuse rhythmic 3.5 Hz spike-and-wave discharges, compatible with childhood absence epilepsy.

**Table 1 jcm-12-02580-t001:** Studies on children with seizures/epilepsy resulting from neonatal hypoglycaemia.

Reference	Epilepsy Onset	Epilepsy Type	Seizure Type	Interictal EEG	Brain MRI	Epilepsy Course
Norden et al., 2001 [61]	-	Symptomatic occipital lobe epilepsy: 4 Pts	FIAS	-	O/PO lesions	Drug-resistant
Caraballo et al., 2004 [54]	Age range: 5 m–10 y	Occipital lobe epilepsy (n = 12)	FAS and FIAS with visual symptoms, FMS, and bilaterally convulsive	Occipital ED (n = 12/12) and slow waves (n = 3/12)	O/PO Lesions (n = 10/12) and normal (n = 2/10)	Variable
		Epileptic encephalopathy (n = 2)	Epileptic spasms, FMS, atonic, and tonic seizures	Hypsarrhythmia and multifocal spikes (n = 2/2)	PO lesions (n = 2/2)	Drug-resistant
Montassir et al., 2009 [50]	Median age: 2 y and 8 m	Occipital lobe epilepsy (n = 6)	FAS and FIAS with visual symptoms	Parieto–occipital ED	O/PO lesions (n = 4/6), hippocampal atrophy (n = 1/6), and normal (n = 1/6)	Drug-resistant in infancy, improvement in older age
Karimzadeh et al., 2010 [46]	Mean age: 20 m	Occipital lobe epilepsy (n = 19)	FIAS with visual symptoms, infantile spasms, and secondary GTC seizures	Posterior ED (n = 14/19) and hypsarrhythmia (n = 2/19). Multifocal ED (n = 1/19) and normal (n = 1/19)	O/PO lesions (n = 15/19), brain atrophy (n = 3/19), and normal (n = 1/19)	Drug-resistant: (n = 8/19); seizure-free or sporadic seizures (n = 11/18)
Kumaran et al., 2010 [62]	Mean age: 6.6 m	Symptomatic infantile Spasms	Spasms in clusters	Classical hypsarrhythmia i (n = 3/5)	Right PO cystic lesion (n = 1/5)	Seizure-free (n = 3/5), drug-resistant (n = 1/5), unknown (n = 1/5)
Fong et al., 2014 [42]	Range: 4 m–5 y	Occipital lobe epilepsy (n = 9)	FAS and FIAS with visual symptoms (n = 9). Infantile spasms (n = 1/9) in 1/9 then followed by occipital seizures	Posterior ED (n = 8/9) and normal (n = 1/9)	O/PO lesions (n = 9/9)	Drug-resistant (n = 3/9), seizure-free, or sporadic seizures (n = 6/9)
		Symptomatic generalized epilepsy: Lennox–Gastaut syndrome (n = 2)	Infantile spasms atonic, tonic, atypical absences, and GTCS	Hypsarrhythmia, multifocal ED, PFA, and slow background	PO lesions (n = 2/2)	Drug-resistant (n = 2/2)
Gataullina et al., 2014 (on patients with IEM) [40]	Range: 1–9 y	Occipital lobe epilepsy 18/21	FAS and FIAS with visual symptoms (n = 13/18), FMS (n = 6/8), atypical absences/drop attacks (n = 5/18), tonic seizures (n = 4/18) myoclonic seizures (n = 3/18), and epileptic spasms (n = 2/18)	-	O/PO lesions (n = 15/18), basal ganglia (n = 3/18), and hippocampal atrophy (n = 1/18)	Drug-resistant (n = 8/18), seizure-free, or sporadic seizures (n = 10/18)
	Range: 2–5 y	Idiopathic epilepsy (n = 3)	FMS, absence seizures, and convulsive seizures	Centrotemporal ED and generalized ED	Normal	Seizure-free
Yang et al., 2016 [63]	Range: 2–10 m	West syndrome (n = 18)/	Infantile Spasms (n = 18/18)	Hypsarrhythmia (n = 18/18)	O/PO lesions 10/18 (n = 10/18)	Follow-up not available
Arhan et al., 2017 [56]	Range: 6 m–15 y	Occipital lobe epilepsy (n = 19/36). West syndrome (n = 8/23) and unclassified epilepsy (n = 9/36)	FAS and FIAS with visual symptoms (n = 19/36), infantile spasms (n = 8/36), and variable (n = 9/36)	Posterior ED (n = 23/36), multifocal ED (n = 10/36), partially overlapping with hypsarrhythmia (n = 8/36), and normal (n = 3/36)	O/PO lesions	Seizure-free (n = 13/36) and drug-resistant (n = 23/36)
Kapoor et al., 2020 [60]	Mean age: 10.3 m	Occipital lobe epilepsy (n = 34), West syndrome (n = 130), LGS (n = 4), and CSWS (n = 2)	FAS and FIAS with visual symptoms (n = 19/170), epileptic spasms (n = 130/170), bilateral convulsive (n = 12/170), myoclonic (n = 6/170), and atonic (n = 3/170)	Hypsarrhythmia (n = 130/170), focal O or TO spikes (n = 18/170), multifocal discharges (n = 8/170), slow spike and wave with bursts of fast rhythm (n = 4/170), CSWS (n = 2/170), and normal (n = 8/170)	O/PO lesions (n = 170/170) and Pulvinar scarring (n = 1/170)	Seizure-free or sporadic seizures (n = 54/170) and drug-resistant (n = 116/170)

m: month/months. y: year/years. CSWS: continuous spikes and waves during sleep. ED: epileptiform discharges; FAS: focal aware seizures; FIAS: focal impaired awareness seizures. FMS: focal motor seizures; GTCS: generalized tonic–clonic seizures. IEM: inborn errors of metabolism. LGS: Lennox–Gastaut syndrome. PFA: paroxysmal fast activity. P: parietal. O: occipital. PO: parieto–occipital. TO: temporo–occipital WM: white matter.

## Data Availability

Additional information on the data may be requested from the co-responsible author.

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
