# Peer review of "Glycaemic Imbalances in Seizures and Epilepsy of Paediatric Age: A Literature Review"

_jcm, 2023, doi:10.3390/jcm12072580_

Round 1
Reviewer 1 Report
The article has been well prepared. It would be better if they mentioned the other clinical manifestations of "Hypoglycemic occipital syndrome" such as visual agnosia and abnormal VEP. On the other hand, it could be considered although hypoglycemia can cause seizures the different types of ketogenic diet including low glycemic index treatment can control the seizures, especially refractory seizures.
The authors should complete the sentence after line number 121 by considering these findings: Other clinical manifestations of this condition that are nominated as Hypoglycemic Occipital Syndrome include visual agnosia
due to the bilateral occipital lobe and you could detect this finding as abnormal VEP. Some patients need rehabilitation in order to improve their visual impairment (43).
Secondly in line 302 before the sentence "Ketogenic diet can be continued indefinitely....." this sentence could be completed by adding these new data;
The Low Glycemic Index Treatment (LGIT) and the Modified Atkins Diet (MAD) are diets recently introduced for refractory epilepsy.MAD and LGIT have an antiepileptic efficacy comparable to KD with fewer side effects.
Author Response
Reviewer 1
- The article has been well prepared. It would be better if they mentioned the other clinical manifestations of "Hypoglycemic occipital syndrome" such as visual agnosia and abnormal VEP. On the other hand, it could be considered although hypoglycemia can cause seizures the different types of ketogenic diet including low glycemic index treatment can control the seizures, especially refractory seizures.
The authors should complete the sentence after line number 121 by considering these findings: Other clinical manifestations of this condition that are nominated as Hypoglycemic Occipital Syndrome include visual agnosia due to the bilateral occipital lobe and you could detect this finding as abnormal VEP. Some patients need rehabilitation in order to improve their visual impairment (43).
Reply: We have now added this interesting neurological observation at line 189 rather than 121 to highlight this specific phenotype in comparison to SELEAS as follows: “The electro-clinical features of the latter can mimic self-limited epilepsy with autonomic symptoms (impaired awareness with paroxysmal eye phenomena such as blinking, clonic movements, tonic deviations, eyelid fluttering), yet ictal vomiting is rare (43,51,63,65,66) (personal observation in Figure 2). These Patients often also exhibit visual loss/agnosia with abnormal visual evoked potentials to compose a complex neurological phenotype defined by Karimzadeh et al. as ‘Hypoglycemia–Occipital Syndrome’. They may need rehabilitation in order to improve their visual impairment (43”).
- Secondly in line 302 before the sentence "Ketogenic diet can be continued indefinitely....." this sentence could be completed by adding these new data; The Low Glycemic Index Treatment (LGIT) and the Modified Atkins Diet (MAD) are diets recently introduced for refractory epilepsy. MAD and LGIT have an antiepileptic efficacy comparable to KD with fewer side effects.
Reply: As suggested we have now modified the paragraph citing additional references as follows: “In Glut1DS, ketogenic diet is first line treatment and should be initiated as early as possible to provide a prompt supplemental supply of metabolic fuel from ketone bodies to the developing brain. The ketogenic diet can also be beneficial for other Patients with drug-resistant epilepsy, irrespectively from the underlying cause, especially in the pediatric age. It can be continued indefinitely but can be poorly tolerated by adolescents and adults. The Low Glycemic Index Treatment (LGIT) and the Modified Atkins Diet (MAD) are diets recently introduced for refractory epilepsy. MAD and LGIT have an antiepileptic efficacy with fewer side effects compare to the ketogenic diets. The MAD uses high fat, low carbohydrate, moderate protein diet to induce ketosis (85). The LGIT stabilizes blood glucose instead of increasing ketone bodies, allowing the intake of a limited amount of carbohydrates (40–60 g/day); the percentage of calories from fat is about 60%, compared with up to 90% in the ketogenic diet (88)”.
Reviewer 2 Report
The authors have done a great effort in exploring the relationship between glycemic and metabolic status and seizure pathophysiology. The manuscript is well written. However, there is always a room for improvement. To my opinion, the manuscript can be more robust and scientific by incorporating the following elements.
1. The reader is sometimes lost due to long continuous and monotonous presentation of information. It would be great if the authors make a thematic diagram/flowchart/figure summarizing the key aspects of the relationship between hypoglycemia and possible development of epilepsy.
2. Epilepsy pathophysiology is deeply related to Increased Glutamate/Aspartate, and decreased GABA/Glycine; and increased Calcium/sodium currents, and decreased Chloride currents. Hence, most of the antiepileptic drugs are based on modulation of the above neurochemicals and ions. There can be few words on this aspect whether there is any relationship between these ion currents/channels/neurochemicals and glycemic status.
3. It can also be discussed whether any of the present anti-diabetic drugs have an off-label use for any subtype of epilepsy.
Author Response
Reviewer 2
The authors have done a great effort in exploring the relationship between glycemic and metabolic status and seizure pathophysiology. The manuscript is well written. However, there is always a room for improvement. To my opinion, the manuscript can be more robust and scientific by incorporating the following elements.
- The reader is sometimes lost due to long continuous and monotonous presentation of information. It would be great if the authors make a thematic diagram/flowchart/figure summarizing the key aspects of the relationship between hypoglycemia and possible development of epilepsy.
Reply: As suggested, to guide the readership amongst different issues we have now drawn a flowchart introduced by the following sentence: “The main issues we will deal with are summarized in Figure 1”. Please note other figures have been renumbered accordingly.
- Epilepsy pathophysiology is deeply related to Increased Glutamate/Aspartate, and decreased GABA/Glycine; and increased Calcium/sodium currents, and decreased Chloride currents. Hence, most of the antiepileptic drugs are based on modulation of the above neurochemicals and ions. There can be few words on this aspect whether there is any relationship between these ion currents/channels/neurochemicals and glycemic status.
Reply: We have now added a brief comment based on additional references on this interesting and very complicated theme as follows: “As a matter of fact, glucose unbalances influence the brittle energy homeostasis of the brain. A disruption of energy availability affects the sodium-potassium pomp and the resting state potential, increases intracellular calcium and reactive oxygen species that promote cell death (17). Hyperglycemia can directly increase neuronal excitability acting on ATP-sensitive potassium channels of hippocampal and neocortical neurons, hypoglycemia depresses GABA levels enhancing excitatory transmission (18,19)”.
- It can also be discussed whether any of the present anti-diabetic drugs have an off-label use for any subtype of epilepsy.
Reply: We have now added a specific paragraph on possible repositioning of antidiabetic drugs for epilepsy treatments based upon recent evidences as follows: “The interplay between DM and epilepsy is further confirmed by recent evidences about the effects of antidiabetic drugs on seizures. Metformin has shown promising results as an antiepileptic agent in many experimental studies. In vivo, it has been successfully employed to reduce seizures in people with Tuberous Sclerosis Complex and Lafora disease (reviewed in (159)). Rosiglitazone can suppress seizures in vitro hampering the presynaptic glutamate release in hippocampal slices (160). These preliminary data may pave the way to repositioning of antidiabetic drugs for epilepsy treatment”.
Reviewer 3 Report
This review covers comprehensive coverage of seizures and epilepsy associated with hypoglycemia and hyperglycemia in neonates and older children. We appreciate this very ambitious review that takes on a major pathophysiology. However, the reviewers agree that the article needs to begin with a clearer definition of hypoglycemia and hyperglycemia, as well as seizures and epilepsy.
Several reviewers' comments are listed below. The authors would like to receive responses on their own opinions throughout the writing of this paper.
(Minor 1)The section is headed Neonatal Seizures and Seizures in Older Children. Specifically, how many days old to days old does neonatal seizures indicate? Also, please state how many years old or older the older infant is, or how many months old or older the infant is.
(Minor 2)Is it possible to clearly indicate what the authors consider to be the normal range of blood glucose levels for each age group? Then, for example, how many hypoglycemia below and for how long does a neonate remain at risk? For older children, how long does hypoglycemia have to last before they are at risk? And please add the same for hyperglycemia.
(Minor 3)It would be clinically beneficial to the reader if you could provide a brief protocol for correction of hypoglycemia, and treatment.
(Minor 4)It would be helpful to add a broad range of comments on organs other than the brain, such as the liver, muscles, hormonal system, etc., where low blood sugar can be a problem.
(Minor 5)The reviewer believes that the most severe form of hypoglycemia-associated seizures and loss of consciousness is metabolic acute encephalopathy. The reviewer may wish to add a perspective on these pediatric blood glucose-related acute encephalopathies.
The reviewers offered some comments, but in general, we rate this review as a valuable and clinically useful article. The reviewers would like to request that the article be resubmitted to JCM with the authors' opinions based on the comments.
Best regards
Dr. Reviewer
Author Response
Reviewer 3
This review covers comprehensive coverage of seizures and epilepsy associated with hypoglycemia and hyperglycemia in neonates and older children. We appreciate this very ambitious review that takes on a major pathophysiology. However, the reviewers agree that the article needs to begin with a clearer definition of hypoglycemia and hyperglycemia, as well as seizures and epilepsy.
Reply: We thank the Reviewer for his//her appreciation and suggestions. The definition of hypoglycemia had already been mentioned in rows 102-105. Likewise, we had already provided a definition for neonatal hyperglycemia in rows 380-383. We have now clarified the parameters to define diabetes mellitus in rows 453-456, with further details in a newly-drafted Supplementary Material file. In the latter, we have also explained definitions of seizures/epilepsy that might be needed for non-neurological readership.
- (Minor 1). The section is headed Neonatal Seizures and Seizures in Older Children. Specifically, how many days old to days old does neonatal seizures indicate? Also, please state how many years old or older the older infant is, or how many months old or older the infant is.
Reply: The cut-off for age of the papers we reviewed is heterogenous. To clarify, we have now included the definition of neonatal age in the main text and those for the different stages of infancy and childhood in the Supplementary Material.
- (Minor 2). Is it possible to clearly indicate what the authors consider to be the normal range of blood glucose levels for each age group? Then, for example, how much hypoglycemia below and for how long does a neonate remain at risk? For older children, how long does hypoglycemia have to last before they are at risk? And please add the same for hyperglycemia.
Reply: There are no specific data about these parameters. As mentioned in rows 102-105, for neonatal hypoglycemia “Although there is no uniform cut-off, a plasma glucose level of 50 mg/dL or less has been considered sufficient to define hypoglycaemia, as many counter-regulatory responses occur at this level”. To provide cut-off for older children is cumbersome. Gandhi nicely reviewed the issue as follows (doi: 10.21037/tp.2017.10.05): “Generally, some main approaches to defining hypoglycemia have included: (I) hypoglycemia relative to presence of clinical manifestations; (II) using population data of values to determine a cutoff based on standard deviation; (III) hypoglycemia at which neurological function is impaired; and (IV) hypoglycemia at which physiologic metabolic and hormonal processes occur (16,17), but these approaches have been problematic for a myriad of reasons. This may suggest that hypoglycemia may be more of a continuum of hormonal abnormalities and clinical manifestations, and a single plasma glucose value has become difficult to associate with neurological outcome, as it could depend on the degree and duration of the hypoglycemia (11,16). In addition, a newborn infant may be asymptomatic, and these clinical symptoms could be attributed to other pathologic conditions (16)”. Likewise, homogenous cut-off for hyperglycemia have not been defined. We have now synthetized these tricky observations in the Conclusions as follows: “To define homogenous cut-offs in children for hypoglycemia/hyperglycemia-induced brain damage is complicated, clinical observation is strikingly important”.
- (Minor 3). It would be clinically beneficial to the reader if you could provide a brief protocol for correction of hypoglycemia, and treatment.
Reply: we have now drafted the goal of treatment for Neonates, Infants, and Children with a Persistent Hypoglycemia Disorder by the Pediatric Endocrine Society as well as treatment advice in the Supplementary Material.
- (Minor 4). It would be helpful to add a broad range of comments on organs other than the brain, such as the liver, muscles, hormonal system, etc., where low blood sugar can be a problem.
Reply: we agree this is an interesting field of investigation. However, we feel that to expand the Review out of the Central Nervous System is much beyond the scope of our study, which is focused on the seizures/epilepsy.
- (Minor 5). The reviewer believes that the most severe form of hypoglycemia-associated seizures and loss of consciousness is metabolic acute encephalopathy. The reviewer may wish to add a perspective on these pediatric blood glucose-related acute encephalopathies.
Reply: We definitely agree. Actually, we had already cited and widely discussed the study by Gataullina et al., see rows 227-237: “Focusing on children with inborn errors of metabolism, Gataullina et al. [40] reported that only the half of patients who suffer seizures during the first hypoglycaemic event (n=90/170; 53%) can experience further seizures later on (n=23/90; 23%). The first hypo-glycaemic seizure could either be self-limited (68%) or very prolonged to establish an overt status epilepticus (32%). Children with status epilepticus at onset would be at risk for fur-ther prolonged episodes, especially triggered by fever. In this series, most children had a symptomatic epilepsy with brain lesions on MRI (86%), whose pattern depended on age at the onset of hypoglycaemia: posterior white matter (0-6 months), basal ganglia (6-22 months), parieto-temporal cortex (>22 months). A minority of patients (14%) developed recurrent hypoglycaemic seizures followed by non-hypoglycemic seizures in spite of normal neuroimaging [40]” and rows 283-289 “Of note, patients with inborn errors of metabolism who suffer episodes of neona-tal-infantile hypoglycaemia can also develop features of idiopathic generalized epilepsy in childhood. The few reported cases all exhibit primary hyperinsulinemic hypoglycae-mia (i.e., excessive insulin production due to known or presumed genetic defects). Gataul-lina et al. [40] reported three patients with episodes of severe recurrent hypoglycaemia du-ring the first 2 years of life, who later developed features of self-limited epilepsy with cen-trotemporal spikes or early-onset absence epilepsy, successfully treated by valproate”.